# 40-Hz Auditory Steady-State Response (ASSR) as a Biomarker of Genetic Defects in the *SHANK3* Gene: A Case Report of 15-Year-Old Girl with a Rare Partial *SHANK3* Duplication

**DOI:** 10.3390/ijms22041898

**Published:** 2021-02-14

**Authors:** Anastasia K. Neklyudova, Galina V. Portnova, Anna B. Rebreikina, Victoria Yu Voinova, Svetlana G. Vorsanova, Ivan Y. Iourov, Olga V. Sysoeva

**Affiliations:** 1Laboratory of Human Higher Nervous Activity, Institute of Higher Nervous Activity and Neurophysiology, Russian Academy of Science, 117485 Moscow, Russia; anastacia.neklyudova@gmail.com (A.K.N.); caviter@list.ru (G.V.P.); anna.rebreikina@gmail.com (A.B.R.); 2Veltischev Research and Clinical Institute for Pediatrics of the Pirogov, Russian National Research Medical University, Ministry of Health of Russian Federation, 125412 Moscow, Russia; vivoinova@yandex.ru (V.Y.V.); svorsanova@mail.ru (S.G.V.); ivan.iourov@gmail.com (I.Y.I.); 3Mental Health Research Center, 117152 Moscow, Russia

**Keywords:** 22q13.3 duplication, auditory steady-state response, ASSR, *SHANK3*, biomarker, auditory event-related potential, ERP, autism spectrum disorders, intellectual disabilities

## Abstract

*SHANK3* encodes a scaffold protein involved in postsynaptic receptor density in glutamatergic synapses, including those in the parvalbumin (PV)+ inhibitory neurons—the key players in the generation of sensory gamma oscillations, such as 40-Hz auditory steady-state response (ASSR). However, 40-Hz ASSR was not studied in relation to SHANK3 functioning. Here, we present a 15-year-old girl (SH01) with previously unreported duplication of the first seven exons of the *SHANK3* gene (22q13.33). SH01’s electroencephalogram (EEG) during 40-Hz click trains of 500 ms duration binaurally presented with inter-trial intervals of 500–800 ms were compared with those from typically developing children (*n* = 32). SH01 was diagnosed with mild mental retardation and learning disabilities (F70.88), dysgraphia, dyslexia, and smaller vocabulary than typically developing (TD) peers. Her clinical phenotype resembled the phenotype of previously described patients with 22q13.33 microduplications (≈30 reported so far). SH01 had mild autistic symptoms but below the threshold for ASD diagnosis and microcephaly. No seizures or MRI abnormalities were reported. While SH01 had relatively preserved auditory event-related potential (ERP) with slightly attenuated P1, her 40-Hz ASSR was totally absent significantly deviating from TD’s ASSR. The absence of 40-Hz ASSR in patients with microduplication, which affected the *SHANK3* gene, indicates deficient temporal resolution of the auditory system, which might underlie language problems and represent a neurophysiological biomarker of *SHANK3* abnormalities.

## 1. Introduction

SH3 and multiple ankyrin repeat domain 3 (*SHANK3*), also known as proline-rich synapse-associated protein 2 (ProSAP2), is a gene that encodes scaffolding proteins that organize postsynaptic density in excitatory synapses [1,2]. This gene is in the 22nd chromosome, 22q13.33 region. Deletion of this region as well as mutations lead to 22q13 Deletion Syndrome also known as Phelan–McDermid Syndrome (PMS) [3,4,5,6,7,8]. In most PMS cases, the *SHANK3* gene is affected that is believed to be the major cause of PMS.

Phelan–McDermid Syndrome (PMS) is a rare neurodevelopmental disorder with about 2000 cases identified so far [6]. However, many PMS cases can go unnoticed, as the diagnosis of PMS is often difficult due to the subtle appearance of the deletion of chromosome 22 and relatively mild physical and nonspecific clinical manifestation of the syndrome. Dysmorphic features in PMS include dysplastic nails, large or prominent ears, long eyelashes, wide nasal bridge, bulbous nose, and sacral dimple [3]. Major dysfunctions in PMS are hypotonia [3], global developmental delay, and severely delayed or absent speech [4]. Autistic traits are also present in most patients with PMS, suggesting PMS as a syndromic form of autism spectrum disorder (ASD) [9,10]. According to a recent meta-analysis, 0.7% of patients with ASD have *SHANK3* mutations, and this number is even higher (2.1%) for ASD patients with moderate to profound intellectual disability [11]. Moreover, altered methylation patterns in *SHANK3* were detected in ≈15% of postmortem autistic brain tissue [12], suggesting even more widespread implication of altered *SHANK3* expression to ASD development through epigenetic influence. Copy-number of variance (CNV) and point mutations of *SHANK3* have been also associated with intellectual disability and schizophrenia [11,13,14,15,16,17,18,19].

Few cases (*n*≈30) of duplication involving the *SHANK3* gene has been described in the literature: among patients with Asperger’s syndrome, attention-deficit hyperactivity disorder (ADHD), bipolar disorder [20], schizophrenia [21], intellectual disabilities, delayed speech and language development [20,21,22,23,24]. While ASD is reported in patients with *SHANK3* duplication, ASD prevalence seems to be smaller than in *SHANK3* deletions or mutations (≈15% vs. >50%). Dysmorphic features of patients with duplications and mutations affected *SHANK3* gene included full lips, slightly upturned nose/anteverted nares, protruding ears, arch-shaped eyebrows. Microcephaly was reported in 15% of reported cases [23]. Resemblances between the cases with *SHANK3* duplication noticed by the researchers points to a distinct 22q13.33 duplication syndrome (for a recent update, see [23,24]). At the same time, the implication of both *SHANK3* deletion and duplication in neurodevelopmental and neuropsychiatric disorders suggests that *SHANK3* gene dosage is essential for correct brain function. However, one must be aware that microduplications does not always mean overexpression of the coded proteins, as an insertion of genetic material within the gene can alter nucleotide sequence and lead to abnormal protein code. Thus, detailed molecular genetic analysis is needed to infer whether the microduplication leads to gain or loss of *SHANK3* functioning.

Several animal models of ASD with deficient *Shank3* gene were developed. Mice with *Shank3* mutations/deletion exhibit ASD-like symptoms including social abnormalities and motor coordination problems [12,14,16,24,25,26,27,28,29,30,31,32]. The transgenic mice with mildly overexpress Shank3 proteins (≈50%) were also created [20,33,34]. These mice display manic-like hyperkinetic behaviors and decreased social interaction; however, unlike *Shank3* knockout mice (KO), *Shank3* transgenic mice did not exhibit repetitive behavior.

Shank3 determines the postsynaptic density of N-methyl-D-aspartate (NMDA) receptors. NMDAR is one of three ionotropic receptors to the main excitatory mediator in the brain: glutamate. Deviation in NMDAR function alters excitation/inhibition balance in neuronal circuitry and associates with autistic-like behavior in patients with ASD as well as in its animal models [26]. It is noteworthy that different *Shank3* mouse lines show similar NMDAR hypofunction [14,16,25,26,27,28,29,30,31].

Recent studies pointed to the abnormalities in inhibitory signaling in *Shank3*-mouse models of ASD. In particular, several studies [16,35] reported the reduced number of synaptic puncta containing parvalbumin (PV) as well as reduced PV expression of the PV-expressing gamma-aminobutyric acid (GABA) interneuron—the most abundant subtype of the inhibitory interneurons, which contribute to the perisomatic inhibition of glutamatergic principal cells. Supporting the implication of reduced inhibition to Shank3 deficits, an enhancer of GABA-mediated inhibitory transmission, clonazepam, normalizes the abnormal network firing pattern in cultured cortical neurons of *Shank3* KO mice [36].

Cortical gamma oscillations (30–100 Hz) are generated in recurrent circuits of excitatory and inhibitory neurons [37,38,39] and reflect the excitatory state of the neural network. While baseline, spontaneous gamma oscillations are studied in humans and animals, high-frequency oscillations are most reliably induced in response to sensory stimuli [40,41,42,43]. The evoked gamma-band activity can be studied with auditory steady-state response (ASSR) [41,42,43]. ASSR refers to the ability of the neural populations to synchronize the timing of neural discharges with the frequency of external periodic auditory stimulation, e.g., click trains or amplitude modulated tone. ASSR is most pronounced in response to 40 Hz stimulation [44], coinciding with an intrinsic resonance frequency of cortical PV+ fast-spiking interneurons [45,46]. This 40-Hz ASSR was recently proposed as a non-invasive biomarker of NMDA receptor function [47,48,49,50]. In mice the pharmacological modulation of NMDAR function by NMDA antagonists such as MK-801 or ketamine suggested an inverse relationship between ASSR and NMDA occupancy [48,51]. Nakao and colleagues [47] demonstrated robust ASSR deficits in the mutant mice with selective elimination of NMDARs from PV+ interneurons in neocortex (Ppp1r2-cre/fGluN1 KO mice), suggesting a causal role of the NMDA receptors on this PV+ interneurons for neural entrainment at 40 Hz. Modeling studies supported this finding, emphasizing the link between NMDAR on PV+ interneurons and 40-Hz ASSR [52,53].

ASSR is reduced in schizophrenia (for meta-analysis see [54]), bipolar disorders [55,56,57,58], and autism spectrum disorders (ASD) [59,60], which are the disorders with implicated GABAergic dysfunctions and altered NMDA signaling. The 40-Hz ASSR deficit occurs in non-psychotic first-degree relatives of patients with schizophrenia [61] and ASD [59], which is consistent with an effect of familial or genetic risk factors. However, recent larger sample studies in children with ASD did not confirm ASSR reduction [62,63]. Such discrepancy might be related to the well-known heterogeneity of the ASD population. Even remarkably similar behavioral manifestations can be caused by different biological underpinnings, e.g., genetic etiology. Thus, examination of ASSR for the patient with known genetic abnormalities, associated with ASD, might be the Rosetta stone for the identification of subgroups of ASD patients based on common molecular–genetic and neurophysiological causes.

Gamma oscillations have been associated with perceptual organization, attention, memory, consciousness, language processing, and motor coordination [64]. The 40-Hz ASSR has been suggested as a candidate mechanism underlying the fast temporal integration and resolution of auditory inputs [41,42,65,66]. In neurotypical controls and elderly population, ASSR was correlated with gap detection threshold [66] and attenuation of speech perception under the presence of noise [65], pointing to the relevance of ASSR to language processing. In patients with schizophrenia, the 40-Hz ASSR positively correlated with the working memory performance [67], attentional functioning [68], and predicted the future global symptomatic outcome (GAF-S2) [69]. Thus, ASSR is linked to the cognitive functioning, which is altered in patients with *SHANK3* abnormalities.

The promising approach in building the causal link between genes and behavior is relating the genetic pathways converging on candidate cellular/molecular processes to the target neurophysiological phenotype. In line with this approach, here, we present the clinical and neurophysiological description of a 15-year-old girl with rare microduplication in 22q13.33, which affects the *SHANK3* gene. The study focused on examination of the 40-Hz ASSR response, which is crucially dependent on PV+ interneurons activity, one of the key targets of the *SHANK3* gene. At the behavioral level, ASSR is thought to reflect temporal integration and resolution of the auditory system and is linked to memory and speech-in-noise processing. Based on this logic, we hypothesize that this girl will have altered ASSR.

## 2. Results

### 2.1. Genetic Information

The girl, further referred as SH01, has normal karyotype (46, XX). Molecular genetics analysis using an SNP array revealed a duplication (size: 16,389 bp) spanning partially *SHANK3*. The duplication affected the first seven exons of the gene (Figure 1).

### 2.2. Phenotyping, Clinical Description

Anamnesis. SH01 was from full-term pregnancy from healthy parents, who were 39 years old at the time of the girl’s birth. Her weight at birth was 3.040 g and length 52 cm, Apgar score was 7/8. Motor milestones were achieved within normal limits with holding her head at 2 months, sat down at 6 months, stood with the support at 10 months, began to walk alone at 11 months. However, language milestones were slightly delayed with the first syllables appeared at 12 months followed by a relatively long time of no phrases. Short sentences appeared at the age of 3. Cognitive development was also delayed, with a lack of interest in books and cartoons until age 3. At about this age, SH01 developed aggressiveness toward peers (e.g., biting) and protest behavior. At kindergarten, she referred her bad behavior to a fictional peer-boy. Aggressive behavior was resolved when she was about 10 years old. Currently, she might have some rare periods of self-aggression (biting) when too angry and unsatisfied. SH01 started normal school together with typically developing (TD) peers, but by the end of primary school, she was referred to specialists due to the problems with dealing with the school program (especially Math). However, she managed to continue the study in the school with TD peers with the support of the specialists and parents. Menstruation was regular and started at 10 years of age.

SH01 took part in our EEG/ERP experiment at age 15.06 years old. Her official diagnoses were mild mental retardation and other deficits of behavior due to other specified causes (F70.88), and organic emotionally labile [asthenic] disorder with unspecified cause (F06.69). Diagnoses were obtained from the recent clinical reports provided by experienced psychiatrists from the Moscow Research Institute of Psychiatry and Scientific and Practical Center for Mental Health of Children and Adolescents, which is a leading Moscow organization for the diagnosis of mental health problems. The report from a psychologist confirmed mild mental retardation by the Wechsler Intelligence Scale for Children (Russian adaptation based on original Wechsler Intelligence Scale for Children [70]): verbal IQ = 71; nonverbal/performance IQ = 64; full-scale IQ = 64. The psychologist also pointed to unstable attention, smaller memory span, a fluctuating but lower speed of performance, quicker tiredness and loss of work efficiency, as well as infantilism, protest behavior, irritability, emotional liability, problems with understanding the social context, lack of self-critique and motivation to overcome difficulties, and a preponderance of recreational entertaining interests over educational and cognitive ones. A speech therapist revealed mild forms of dysgraphia and dyslexia.

Parents’ major concerns at age 15 were learning disabilities, behavioral disorders, and irritability. The girl was sociable, and her mild cognitive impairment was hardly noticeable in daily routines. Her mild speech underdevelopment manifested in rare problems to pronounce long and complex words and smaller vocabulary than typically developing (TD) peers. She attended the 9th grade of normal middle school that required great efforts from her parents. While she hardly managed to make any homework by herself, she was considering continuing her education in high school, pointing to the lack of adequate self-assessment. Among her interests was performing in a school theater. She used her right hand to write and to eat. At EEG/ERP examination, she showed infantile childish behavior, demanding attention of others and especially her mom, which was not typical for her age (e.g., she asked her mom to stay with her in the experimental room).

Physical parameters at the age of 15: 163 cm (50–75 percentile), 50 kg (50–75 percentile), head circumference of 51.5 cm (lower than 3rd percentile). Facial phenotype included elongated face, protruding auricles. There were short 5th fingers on the hands, a sandal gap. The girl had mild scoliosis, valgus deformity of the knee joints, and planovalgus feet.

Autistic characteristics as assessed at age 15. SH01′s T-scores on social responsiveness scale (SRS) equals 63, which referred to mild autistic symptoms [71], while neither the Autism Diagnostic Interview-Revised [72] (ADI-R, with subscale social interaction A-4 scores, Communication and language B-2, repetitive and restricted behavior C-1, early developmental problems, 1-36 months, D-1) nor psychiatric assessment support the ASD diagnosis.

Magnetic resonance imaging (MRI) at age 15: The hemispheres of the brain were symmetrical. No focal changes in the intensity of the MR signal from the substance of the brain, cerebellum, or brain stem were found. Differentiation into cortical and medullary substances was expressed satisfactorily. The lateral ventricles were symmetrical, not dilated. The hind horns were deepened. The cerebellum was typically located. The pituitary gland was not changed with preserved structure. The adeno- and neurohypophysis was clearly differentiated. Chiasma did not change. The optic nerves were clearly visible. The median structures were not displaced. The craniovertebral junction was not changed.

Other laboratory examinations at age 15. Echocardiography revealed ectopic chords and trabeculae in the left ventricular cavity, mitral valve prolapse with 1+ regurgitation, tricuspid valve prolapse with 1.5+ regurgitation. Ultrasound examination showed bilateral nephroptosis. X-ray showed a short fifth finger metacarpal bone of the left hand. Pulmonary examination revealed moderate bronchial asthma, atopic, with polyvalent sensitization.

Medications at age 15: SH01 took phenibut, 250 mg three times a day to control behavior and levothyroxine (L-thyroxine) 50 mL to treat her asthma (diagnosis J45.0—Predominantly allergic asthma).

### 2.3. Clinical EEG

The voltage of EEG activity was in accordance with the healthy peers’ EEG voltage; significant asymmetry of the background EEG was not detected. EEG recordings with eyes closed demonstrated normal background EEG with dominate alpha rhythm (Figure 2a). In 2018, it had an amplitude of 107 µV and 87 µV (maximal and mean, respectively) and a frequency of 9.1 Hz in the left hemisphere and amplitude of 104 µV and 69 µV (maximal and mean, respectively) and frequency 9.3 Hz in the right hemisphere. In 2020, the dominate alpha rhythm in the eyes closed condition had an amplitude of 93 µV and 67 µV (maximal and mean, respectively) and frequency of 9.5 Hz in the left hemisphere and amplitude of 94 µV and 63 µV (maximal and mean, respectively) and frequency of 9.7 Hz in the right hemisphere. The abnormalities of the background EEG (Figure 2b) could be described as intermittent theta, slowing (3.5–5.5 Hz and 80–140 µV) in the right hemisphere in 2018; in 2020, abnormalities of the background EEG could be described as sporadic spike and polyspike discharges (100–150 µV) arising from the right centrotemporal region.

### 2.4. ASSR/Auditory ERP

SH01′s 40-Hz ASSR and auditory ERP were compared with two control groups of typically developing (TD) children: the first one (“old”, *n* = 13, seven females, mean age 16.04 (SD = 1.9), ranged 12–18) was age-matched with our patient SH01 (age = 15.06). The second subgroup (“young”) consisted of 19 participants (14 female, five male) with an average age of 7.8 (SD = 2.6), ranged 3–12. Comparison groups of different ages were selected to examine if the suggested alternation in SH01′s neurophysiological responses to sounds might be linked to the developmental delay in brain maturation (as changes of auditory ERP and ASSR with age are known, see Section 3) or represent more general phenomena. Table 1 summarizes the results.

The 40-Hz ASSR was clearly identified in the TD groups and was dominant at frontal sites (Figure 3 and Figure 4). Consistent with previous reports, 40-Hz ASSR peaked about 200 ms post-train onset and persisted over the whole period of stimuli presentation in all TD participants, which were significantly higher in the older control group than in the younger one (*t* (30) = 2.362, *p* = 0.025), as can be also seen in Figure 5, which represents the individual 40-Hz ASSR values averaged over the whole period of stimuli presentation. At the same time, 40-Hz ASSR were totally absent in SH01 (Figure 3), being significantly smaller compared to any of the TD groups (old vs. SH01: *t* (12) = 9.6602, *p* < 0.0001; young vs. SH01: *t* (18) = 5.684, *p* < 0.0001). Moreover, there were no TD participant in the old, age-matched group who had 40-Hz ASSR value below that of SH01 (minimum value in the TD old group being 0.053 µV, SH01′s ASSR = −0.015 µV), suggesting a very robust effect (Figure 4, Figure A1 in Appendix A).

Figure 6 represents the auditory event-related potentials to the same stimuli that fail to elicit 40-Hz ASSR in SH01. Despite such a drastic alteration in ASSR response, auditory ERP in SH01 were much more similar to that of TD groups (Figure 6, Figure A2 for individual ERPs). The old TD group was characterized by prominent P1, N1, and N2 components, which were registered after the 40-Hz train onset. SH01′s ERP generally resembled that of the old TD ERPs, with only SH01′s P1 components being significantly smaller than that in her age-matched group (t (12) = 3.484, *p* = 0.005), while N1 (t (12) = 1.864, *p* = 0.087) and P2 (t (12) = −2.099, *p* = 0.058) were unremarkable as represented in Figure 6 and Figure 7. For the peak values of major ERP components, see Table 1. As for younger TD participants, their ERPs was characterized by the absence of a clear N1 response, corresponding with well-known developmental change in the ERPs structure (old TD vs. young TD: *t* (30) = −3.524, *p* = 0.0014). For all components, the SH01′s ERPs differed from the young groups (P1: *t* (18) = 5.683, *p* < 0.0001; N1: *t* (18) = 5.863, *p* < 0.0001; P2: *t* (18) = 2.554, *p* = 0.02). Thus, the auditory ERP in SH01 was more similar to her peers than to the young control group, pointing to a generally preserved development of auditory ERP structure.

## 3. Discussion

Our report presents a new patient with unique duplication of the first seven exons of the *SHANK3* gene, adding one more case to the about 30 patients with 22q13.3 duplications described in previous studies [23]. For the first time, we describe the neurophysiological phenotype of a patient with 22q13.3 duplications. The major focus of our study was on the 40-Hz ASSR, a brain response to high-frequency auditory stimulation, which is thought to underlie temporal binding and speech-in-noise processing [65]. This choice was motivated by the studies that reported 40-Hz ASSR as a biomarker of NMDAR density and PV+ interneurons functioning, as they are dependent on *SHANK3* gene activity [48,49,50,51]. Here, we report a striking absence of 40-Hz ASSR in SH01, collaborating our initial hypothesis. Below, we discuss our findings in more detail.

The clinical phenotype of SH01 resembles that described for few patients with 22q13.3 microduplication (*n* = 29, [23], Table 2), although clinical features in the 22q13.3 duplication syndromes show great variability. Among common features are intellectual disabilities (*n* = 15), attention deficits (*n* = 5), and language problems (*n* = 11). Physical dysmorphic features have been also reported in these patients previously, including sandal gap (*n* = 1) and protruding or low-set deformed ears (*n* = 3), microcephaly (*n* = 5). One previously described patient with 22q13.3 microduplication [21] shared with our patient irritability and scoliosis, as well as mild mental retardation and attention deficits. It is noteworthy that a girl showed normal development until 13 years old but later was diagnosed with borderline intellectual functioning and disorganized schizophrenia. At the same time, unlike few patients with 22q13.3 microduplication who were diagnosed with autism spectrum disorders (*n* = 5) and epilepsy (*n* = 4), our patient SH01 does not have epilepsy, only some minor epileptiform activity in EEG, and does not have enough symptoms to get a diagnosis of autism spectrum disorder, while her SRS score suggested some autistic features. SH01 phenotype was also compared to the more studied 22q13.3 deletion syndrome. SH01 shared with patients of this syndrome intellectual disabilities and language problems, as well as autistic features, although their manifestations are milder in SH01 [3,73,74,75]. Among the dysmorphic features reported in patients with PMS, SH01 also has an elongated skull. Thus, the clinical description of SH01 contains both common and distinct features with patients with different types of abnormalities affecting *SHANK3*, while it more resembles those with *SHANK3* microduplications, pointing to a partially distinct phenotype of 22q13.3 duplication and deletion. For convenience, Table 2 shows the prevalence of individual clinical features in patients with *SHANK3* duplication and deletion.

Our study indicates a general preservation of auditory ERP in SH01 with the pronounced N1-P2 response, which is typical for TD teens. At the same time, the P1 component that usually decreases with age [83,84] is not evident in SH01, with amplitude within P1 latency being significantly smaller in SH01 not only as compared to the younger cohort but also as compared to the age-matched control group. Unfortunately, we are not aware of any ERP study conducted in patients with the 22q13.3 microduplication. Thus, we compare our results with those obtained in patients with point mutations or deletion in 22q13.3 (PMS). Consistent with our finding of reduced P1 response to auditory stimuli, Reese and colleagues [85] found a reduction of P50 in response to the repeated tone in patients with PMS. It is noteworthy that the reduction was significant only for the female participants. Thus, there might be some common deficits in the early stage of auditory processing in the auditory cortex in patients with abnormalities related to the *SHANK3* gene. The decrease in the early component of visual ERP to checkerboard stimuli, which was registered within the same latency range, 50-75 ms post-stimulus, was also reported in PMS [86,87], pointing to the fact that neurophysiological abnormalities occur in PMS at the early stages of sensory processing regardless of the modality of stimulation. Whether such deficits also spread to the visual system in our patient was not studied. It is noteworthy that the attenuation of P1 in response to auditory stimuli was also reported in patients with idiopathic autism [88,89,90], linking these behavioral and neurophysiological abnormalities.

As for the later components, patients with PMS showed a reduction of P2 component in response to the repeated tones [85,91] as well as a decrease in the latency of N250 in response to oddball stimuli [92]. In our study, neither P2 nor N250 were affected, and P2 even tended to be larger in SH01 than in the age-matched controls. Such discrepancy might indicate different neurophysiological phenotypes for 22q13.3 duplication/deletion or just be related to a methodological difference between the studies.

The focus of our study was 40-Hz ASSR, as we hypothesize its abnormality in our patient based on previous literature. Indeed, we found a striking absence of 40-Hz ASSR in SH01. Considering relatively normal auditory ERP in SH01, such finding points to specific deficits in following high-frequency auditory signals. An absence of 40-Hz ASSR might underlie a disruption of temporal integration and binding mechanism in audition that is linked with PV+ interneurons functioning [41,42]. As 40-Hz ASSR was correlated with speech-in-noise perception [65], an absence of ASSR might be related to speech decoding. At the same time, 40-Hz ASSR seems to reflect not a primary mechanism for speech comprehension, as the total absence of 40-Hz ASSR does not prevent SH01 from learning language and being fluent in everyday life. Rather, 40-Hz ASSR reflects some modulatory mechanism that helps to differentiate speech sounds, making it easier to learn and communicate. Abnormalities in such modulatory mechanism can cause low vocabulary and some complex words’ pronunciation problems, as well as mild dysgraphia and dyslexia, as observed in SH01. Further studies are needed to fully examine SH01′s phonematic abilities, which are related to speech perception, to shed light on the particular process disrupted with the absence of 40-Hz ASSR related to *SHANK3* abnormality.

While ASSR is modulated by age [44,93,94], our 15-year-old patient’s 40-Hz ASSR was significantly smaller not only as compared to age-matched control group but also in relation to data obtained in children aged 3–12 years old. Thus, her 40-Hz ASSR deficiency is hard to explain by the developmental delay, or such a delay should be very profound with SH01′s ASSR corresponding to that from TD children under 5–8 years old [95,96].

While 40-Hz ASSR was previously studied neither in patients with 22q13.3 deletion/duplication nor their animal models, Engineer and colleagues [97] observed a drastic attenuation of neuronal firing rate in response to rapidly presented sounds in the *Shank3*-deficient rat model. These authors showed that the number of driven spikes evoked by noise bursts and speech sounds as well as the spontaneous firing rate were significantly weaker in *Shank3*-deficient rats compared to control rats. In relation to our results, the effect was most pronounced when the stimuli were presented with short inter-stimulus intervals below 100 ms, especially in the primary auditory cortex and anterior auditory field. Taken together, the results point to the problems of following the rapidly presented auditory signal as general phenomena, which is related to *SHANK3* abnormalities. This might indicate that the gain and loss of *SHANK3* function share a common neurophysiological phenotype. It might also point to the potential loss-of-function effect of microduplication within the *SHANK3* gene. More detailed molecular genetic analysis and modeling might help resolve these alternatives.

Our study has implications to the heterogeneous population of idiopathic ASD with a significant percentage of its cases related to *SHANK3* abnormalities and having language problems. ASSR being easy to assess as a non-invasive index of functioning of PV+ neurons and NMDAR signaling, stemming from *SHANK3* abnormalities, can help segregate the ASD population based on neurophysiological and molecular genetic underpinnings.

Our study is not without limitations. First, it is just a case report of one patient’s data. While SH01 is an incredibly unique patient, more studies are needed to confirm our observations. As there have been about 30 patients described so far, our study aims to promote the ASSR paradigm among other researchers and clinicians, inviting them to run 40-Hz ASSR in other patients with the 22q13.3 duplication identified world-wide. These studies are an important step toward the validation of this neurophysiological biomarker.

SH01 also took phenibut as regular medicine. As this drug is a GABA-receptor agonist, it might potentially influence ASSR. To rule out such an explanation, we compared SH01 with a kid without known genetic abnormalities, who also took Phenibut for migraine treatment. Such control kid exhibits typically pronounced 40-Hz ASSR (Appendix B, Figure A3). At the same time, more research on a larger sample is needed to examine the effects of phenibut on ASSR.

One may also relate absent 40-Hz ASSR to the hearing problems, arousal level, or attention deficits, as some researchers found ASSR modulation by these factors [44,98,99]. However, normal auditory ERP in response to the same stimuli rule out these explanations, as e.g., an N1 component was also shown to be modulated by attention and stimuli-subjective intensity [100]. Moreover, the P2 component, that was reported to be attenuated in participants with moderate to severe sensorineural hearing loss [101] even tended to be larger in SH01, pointing to an increased rather than decreased auditory sensitivity in SH01.

## 4. Materials and Methods

### 4.1. Participants

Thirty-two typically developing (TD) children were recruited from the local community to take part in a study as a comparison group. According to their parents or guardians, they did not have a history of neuropsychiatric conditions and had normal or corrected to normal vision and hearing. Except for one participant (this case is described in Appendix B), none of the participants reported to be taking any medicines. TD participants were divided into two subgroups by age (Appendix C, Table A1). The first one (“old”) was age-matched with the participant SH01 (age = 15.06). It consisted of 13 people (7 female, 6 male) with an average age of 16.04 (SD = 1.9). The second subgroup “young”) consisted of 19 participants (14 female, 5 male) with an average age of 7.8 (SD = 2.6).

Almost all participants’ guardians filled in the Russian translation of the Social Responsiveness Scale, second edition (SRS-2) [71], the school age version for the ”young” group and the school age or adult version for the “old” group. Threshold values for any social behavior deficiencies are 58 T-scores for males and 63 T-scores for females. None of the participants from the “old” group exceeded the threshold value (range 16–56, mean = 37, SD = 13), and only one participant from the “young” group had a greater value (range 11–62, mean = 27, SD = 14). Six participants did not have SRS values. More detailed characteristics of comparison samples are presented in Table A1.

The SH01 patient underwent a full clinical assessment at the Research Clinical Institute of Pediatrics by experienced clinicians. In addition, Autism Diagnostic Interview–Revised [72], an investigator-based semi-structured instrument, was administered by a trained interviewer to SH01′s mom. It was used to assess autistic traits in SH01.

The study was approved by the local ethics committee of the Institute of Higher Nervous Activity and Neurophysiology, Russian Academy of Sciences, protocol number 3, date of approval July, 10th, 2020, and was conducted following the ethical principles regarding human experimentation (Helsinki Declaration). All children provided their verbal consent to participate in the study and were informed about their right to withdraw from the study at any time during the testing. Written informed consent was also obtained from a parent/guardian of each child.

### 4.2. EEG Recording

Electroencephalographic data were recorded using the NeuroTravel system with 32-scalp electrodes arranged according to the international 10–10 system. Ear lobe electrodes were used as reference, and the grounding electrode was placed centrally. For clinical EEG, periods of resting activity were recorded as well as a test on closing and opening the eyes.

### 4.3. Stimuli and Paradigm

The ASSR paradigm consisted of 40-Hz click train stimuli, which were presented binaurally through foam insert earphones for 500 ms at 80 dB sound pressure level. Inter-trial intervals varied from 500 to 800 ms. The total number of trials was 150, and the duration of the whole paradigm was around three minutes. Stimuli were presented via Presentation software (NeuroBehavioral Systems, Albany, CA, USA). During the experiment, participants were sitting in a dimmed room and watching a silent video of their choice.

### 4.4. Data Analysis

EEG analysis was performed using MATLAB (Version—b, The MathWorks, Natick, MA, USA), Fieldtrip software (https://www.fieldtriptoolbox.org/, [102]), as well as customized scripts. Peak values were compared using two-tailed Student’s t-test for independent groups.

#### 4.4.1. ASSR Analysis

First, the raw data were segmented into epochs with an interval of 200 ms before the stimulus and 800 ms after it. Then, the signal was filtered at a frequency of 35–45 Hz, and trials with amplitude within 3 STD of the mean were averaged. The mean number of selected trials was 97 ± 34 for the “old” group and 80 ± 17 for the “young” group. There were 182 good trials for SH01. To better characterize 40-Hz ASSR, we extracted the envelope of the signal using the Hilbert transform. The absolute value of this linear integral transformation reflects the envelope of the grand-average waveform (see Figure A4). Baseline correction for the 200–0 ms was applied. These steps were conducted for all participants, including the patient with microduplication affecting *Shank3*, SH01. For further analysis, we chose the Fz electrode, since according to topographic data (see Figure 3), ASSR has a maximum response near Fz. It is also consistent with the literature, which reports that ASSR is most pronounced in this site [103,104]. Then, we averaged the values of the envelope curve after Hilbert transform in the Fz electrode over the whole period of stimuli presentation (0–500 ms) and compared the results of SH01 with the average values of each comparison group.

#### 4.4.2. ERP Analysis

ERP for auditory stimuli were created with filtering band-pass 1–30 Hz using the Fieldtrip function for all participants. The averaging epoch was the same as in ASSR analysis: 200 ms before the stimulus and 800 ms after it, but for later analysis, we focused on the period of –200–400 ms. Only trials with an amplitude within 3 STD of the mean were averaged. The mean number of good trials was 99 ± 43 for the old group, 96 ± 18 for the young group, and 69 for SH01. Then, we calculated ERPs peak values for all participants. The timeframes for each component were chosen based on grand-averaged peak latencies (P1: 50–80 ms; N1: 80–120 ms; P2: 130–160 ms).

### 4.5. Molecular Genetic Analysis

Molecular genetic analysis of SH01 was performed by CytoScan HD Arrays (Affymetrix, Santa Clara, CA, USA), which consists of about 2.7 million markers (resolution: >1 kbp). The results were visualized by the Affymetrix ChAS (Chromosome Analysis Suite) CytoScan^®^ HD Array software (reference sequence GRCh37/hg19). The procedures were earlier described in detail [105,106]. All the genomic variations uncovered by the molecular genetic analysis were analyzed using a panel of bioinformatic techniques targeting the phenotypic outcome of each variation. The procedure was described previously in a step-by-step manner [107,108].

## 5. Conclusions

Our study demonstrates a link between duplication of the first seven exons of the *SHANK3* gene and alteration of brain response to high-frequency auditory input, 40-Hz ASSR: a neurophysiological phenotype believed to be mediated by hypofunctional NMDA receptor signaling on the parvalbumin (PV)+ inhibitory neurons, which depends on SHANK3 abnormality. As reported in our manuscript, the absence of 40-Hz ASSR in the patient with microduplication that affected *SHANK3* gene points to a deficient temporal resolution of this patient’s auditory system, which might underlie the language problems observed in our patient as well as in many patients with abnormal functioning of the *SHANK3* gene.

## Figures and Tables

**Figure 1 ijms-22-01898-f001:**
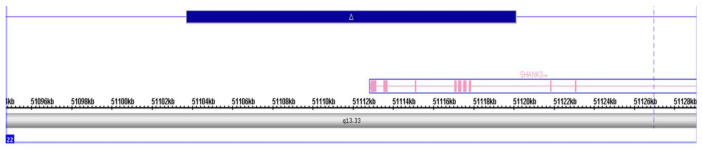
SNP array analysis demonstrating the duplication affecting the first seven exons of the *SHANK3* gene.

**Figure 2 ijms-22-01898-f002:**
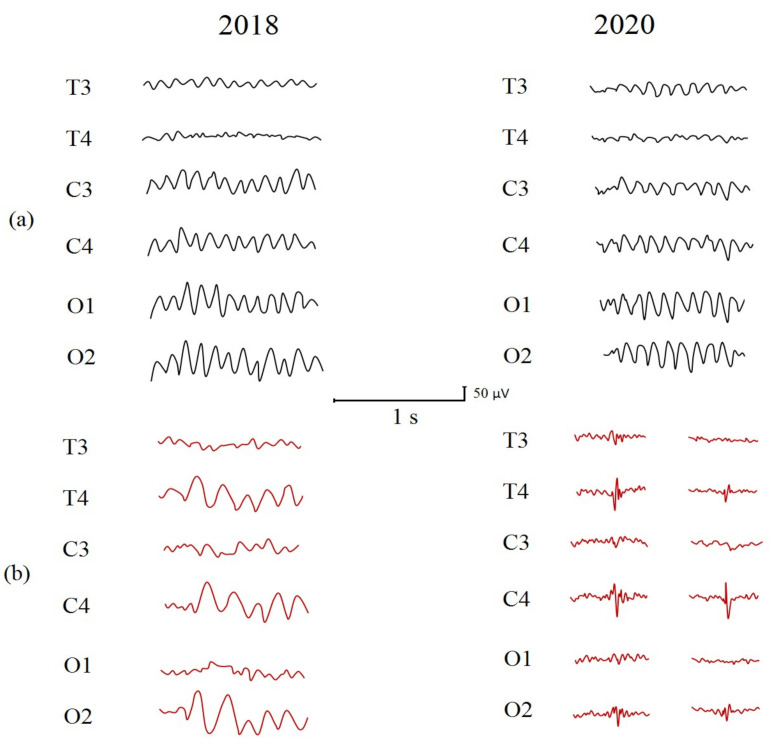
(**a**) Dominate alpha rhythm with eyes closed, (**b**) the abnormalities of the background EEG.

**Figure 3 ijms-22-01898-f003:**
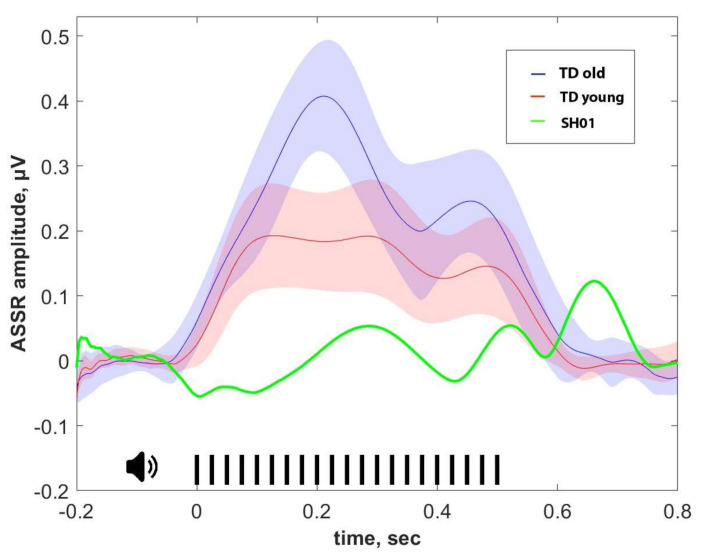
Envelope curve of 40-Hz ASSR obtained after Hilbert transform from electrode Fz. The ASSR of SH01 is shown in green, that of the young group of typically developing children (TD young) is in red, and that of the old, age-matched to SH01 group (TD old) is in blue. Opaque blue and red shading illustrate the 95% confidence interval. The time of stimulus presentation is 0.

**Figure 4 ijms-22-01898-f004:**
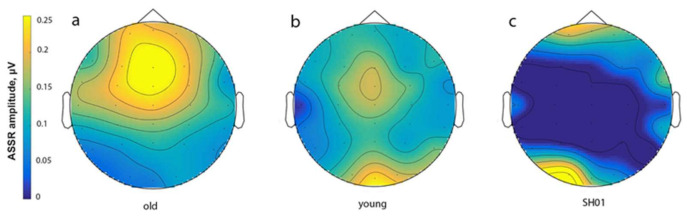
Topographic map of 40-Hz ASSR amplitude averaged over the period of 0–500 ms. The “old” group is represented in (**a**), the “young” group is represented in (**b**), and the values of SH01 are represented in (**c**).

**Figure 5 ijms-22-01898-f005:**
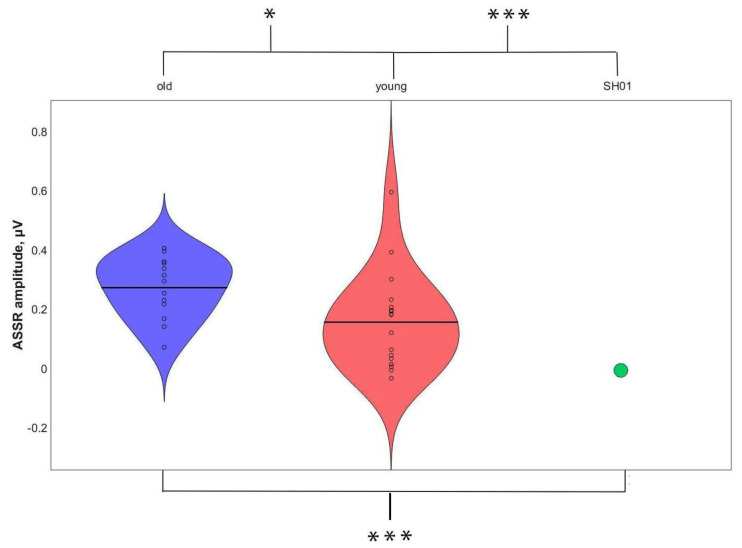
Individual values of 40-Hz ASSR across the groups (Fz electrode). The old TD group’s values are shown in the first column, the young TD group’s values are shown in the second, and SH01’s values are shown in the third. (* shows significant differences *p* < 0.05, *** shows significant differences *p* < 0.0001).

**Figure 6 ijms-22-01898-f006:**
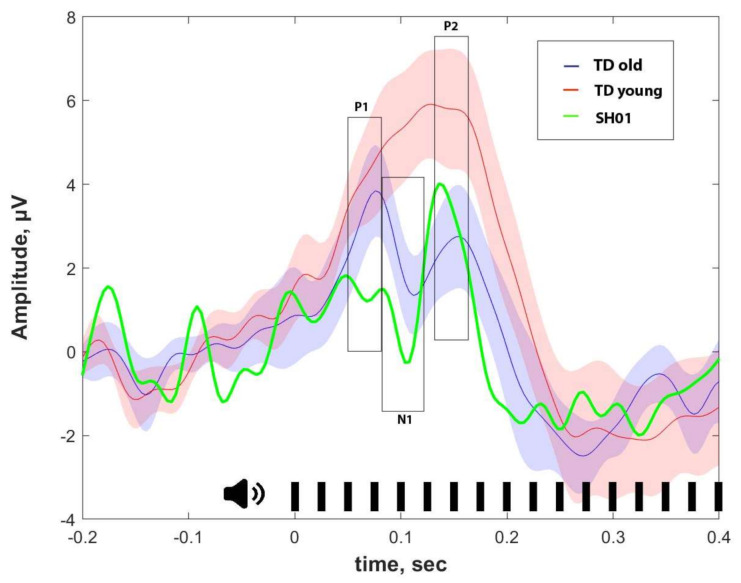
Auditory event-related potentials (ERPs), in Fz electrode with SH01 shown in green, the younger TD group (TD young) is shown in red, and the old, age-matched with SH01 control group (TD old) is shown in blue. Opaque blue and red shading illustrate 95% confidence interval. The time of stimulus presentation is 0. Time windows for P1 (50–80 ms), N1 (80–120 ms), and P2 (130–160 ms) are shown in rectangles.

**Figure 7 ijms-22-01898-f007:**
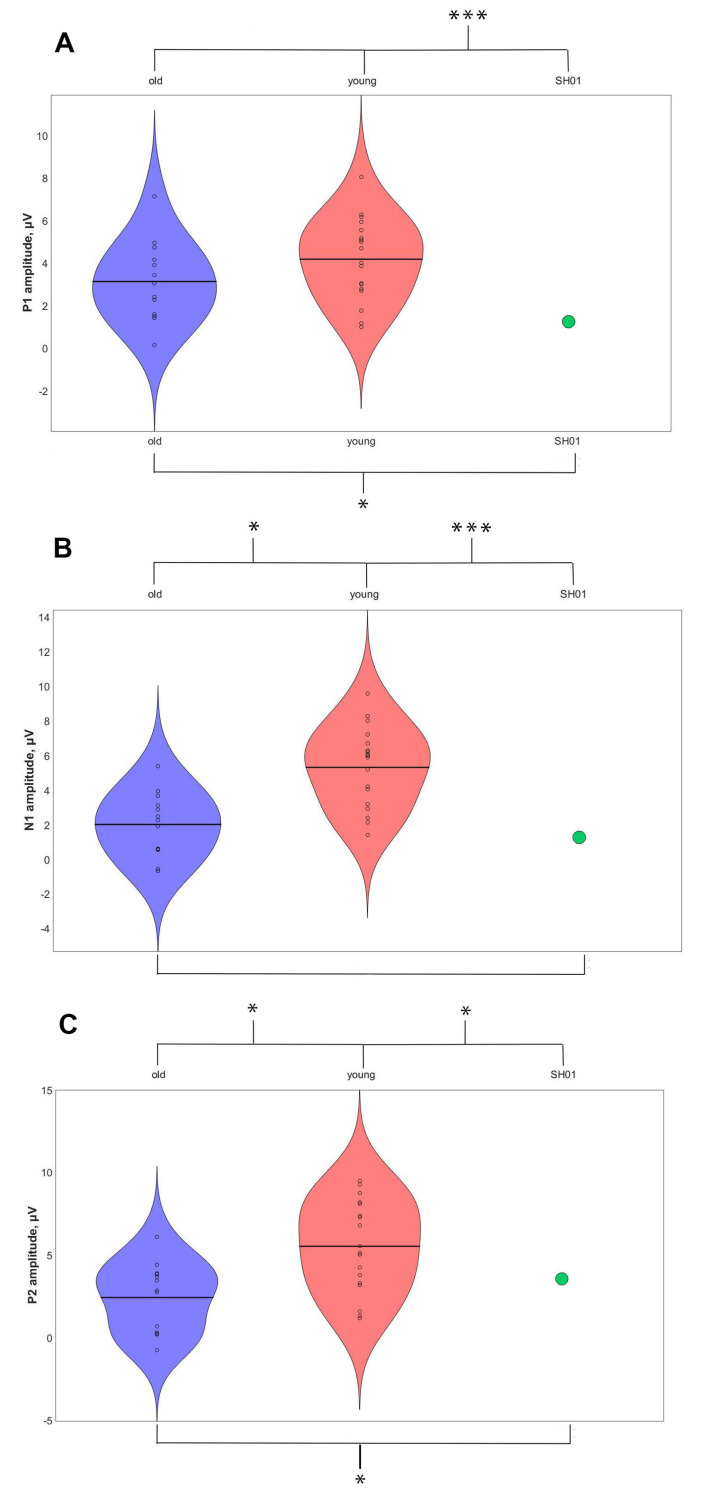
ERPs components across groups. P1 component is shown in (**a**), N1 is shown in (**b**), P2 is shown in (**c**). The old typically developing (TD) group’s values are shown in the first column, the young TD’s values are shown in a second, and SH01’s value is shown in the third (* shows significant differences *p* < 0.05, *** shows significant differences *p* < 0.0001).

**Table 1 ijms-22-01898-t001:** Amplitudes of 40-Hz auditory steady-state response (ASSR) and event-related potential (ERP) components (mean ± STD) for two comparison groups of typically developing children and SH01 in Fz electrode.

	ASSR, μV	P1, μV	N1, μV	P2, μV
SH01 (15.1 y o)	−0.002	1.359	1.102	3.670
Old group (16.1 ± 1.9 y o)	0.274 ± 0.103	3.159 ± 2.017	1.829 ± 1.885	2.332 ± 2.241
Young group (7.8 ± 2.6 y o)	0.158 ± 0.155	4.267 ± 1.886	6.059 ± 3.801	6.455 ± 4.754

**Table 2 ijms-22-01898-t002:** Clinical phenotype of patients with 22q13.3 duplication and 22q13.3 deletion/mutations. In addition to individual cases of patients with 22q13.3 duplication (that include *SHANK3* gene), the last two lines show the overall occurrence (in percentage) of the symptoms in patients with 22q13.3 duplication (based on [21] and our own review of individual cases reported previously) and 22q13.3 deletion/mutations (taken from previous reviews). Our subject, SH01, is also included for comparison. As patients of different ages are described, both IQ and developmental quotient (DQ) are used to characterize mental retardation. +/− reflects the presence/absence of the symptom.

	Language Problem	Mental Retardation	Autism	Microcephaly	Seizures	Attention Deficit	Affective/ Psychiatric Symptoms	Physical Abnormalities/ Dysmorphism
SH01, our patient	mild dyslexia, dysgraphia	mild, IQ 64	SRS = 63, no diagnosis	+	−	+	irritability, aggressiveness	elongated skull, protruding auricles, sandal gap
Johannessen et al. (2019) [23]	n/a	mild	n/a	n/a	+	+	bipolar disease	Tourette syndrome (motor tics), mild dysmorphism
Han et al. (2013) [20] Patient 1	n/a	mild	n/a	n/a	+	+	destructive behavior	dysmorphism
Han et al. (2013) [20] Patient 2	n/a	mild	n/a	n/a	+	+	bipolar disorder	−
Chen et al. (2017) [76]	language delay, echolalia	mild, IQ 62	autistic traits, no diagnosis	n/a	n/a	+	irritability	−
Ujfalusi et al. (2020) [24] Patient 1	−	mild, IQ 72	n/a	−	−	n/a	−	dysmorphism
Ujfalusi et al. (2020) [24] Patient 2	dyslexia, dysgraphia	mild, IQ 79	n/a	−	+	−	bipolar disorder, temper tantrums	dysmorphism
Okamoto et al. (2007) [77] Patient 1	language delay	moderate, DQ 40	−	−	-	−	−	hypotonia, dysmorphism
Okamoto et al. (2007) [77] Patient 2	language delay	moderate, DQ 46	−	−	−	−	−	hypotonia, dysmorphism
Failla et al. (2007) [21]	incoherent speech	mild, IQ 73	−	+	−	+	schizophrenia, irritability, aggressiveness	dysmorphism
Destrée et al. (2011) [78] Patient 1	language delay	mild	n/a	+	n/a	n/a	n/a	dysmorphism, growth retardation
Destrée et al. (2011) [78] Patient 2	language delay	mild	n/a	+	n/a	n/a	n/a	dysmorphism, growth retardation
Duplication (*n* = 31), %	35%	80% (from mild to severe)	19%	17%	17%	16%	bipolar disorder –4% psychosis−7%	Dysmorphism-54%
Deletion/PMS, %	100% (no speech in 50%) [79]	96% (with profound in 53%) [74]	31–84% [74,79]	6–14% [80]	63% [81]	11% [74]	bipolar disorder –54% psychosis–12% irritability–36% [82]	68–93% dysmorphism [74]

## Data Availability

Data available on request due to restrictions.

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
