# Peer review of "40-Hz Auditory Steady-State Response (ASSR) as a Biomarker of Genetic Defects in the *SHANK3* Gene: A Case Report of 15-Year-Old Girl with a Rare Partial *SHANK3* Duplication"

_ijms, 2021, doi:10.3390/ijms22041898_

Round 1

Reviewer 1 Report

Detailed and well performed research case report.

Results are clearly presented and discussion is robust.

The authors also performed an intensive search for comparing their case with the current literature.

Author Response

We thank the reviewer for the high evaluation of our work.

Reviewer 2 Report

Thank you for the opportunity to review this manuscript. The authors present a case study of an individual with a microduplication on the SHANK3 gene. These types of studies are  important for deeply understanding the phenotype and how that can help us understand both underlying neurology and genetics of other disorders. The level of detail in this manuscript is nice and the addition of the EEG findings offers novel information. My biggest concern with the manuscript was in the presentation of the behavioral phenotype. These are  detailed below. 

  • Lines 40-44: Where do the authors get the information on the prevalence and symtompatology. There are at least 2000 individuals with PMS in the worldwide registry and we know that that would be a large underestimate given the likely prevalence of 1:8000-15,000. Further, though the deletion may be difficult to detect genetically, there are lots of behavioral symptoms evidenced from an early age, typically infancy. At the very least these assertions should be referenced.
  • Line 62, I would recommend being more specific rather than saying ‘mental problems’
  • Line 120, missing a ‘the’ before Rosetta Stone
  • Section 2.2 – The diagnosis information is very vague. What are ‘other deficits of behavior due to unspecified causes’? This seems out of place with actual WHO diagnoses also provided. Where did the diagnoses come from? Did the research team do them or were they provided from past reports? The paragraph also becomes redundant with the parents complaining of intellectual disability and speech underdevelopment. Since the case study is the primary focus of this paper, I would recommend reworking so that this section is more flushed out.
  • Line 157, the ‘has’ is not necessary before ‘…started at 10 years of age’
  • What is meant by ‘on examination she showed infantile behavior, too dependent on her mom’? I would also flush this out as a behavioral observation section so we can get a better picture of her as this also does not really align with her ability to succeed in a mainstream classroom
  • Line 160, this paragraph also seems out of place. If the family history is a heading, I would denote it as such.
  • To give context to the motor milestones, particularly because of noted low tone at least in those with the deletion, I would recommend starting with motor milestones were achieved within normal limits with holding her head at ….. This would give the reader context to the fact that these were within typical limits. Similarly, I might say that language milestones however were delayed with …
  • With respect to cognitive, I would then just say cognitive or precognitive milestones were also delayed, with lack of interest in books until and learning was also delayed (or whatever is meant by moderately retarded in mental development).
  • With regard to the aggression, I would say that it resolved or something else rather than eliminated but also would be nice to know if she had treatments for this.
  • For the ASD behaviors, was the SRS current? Similar with the ADI?
  • Line 176, assuming that the authors mean the Wechsler scale, though it would be good to know which version
  • Section 2.4 – what is the rationale for the old and young comparison groups? The young group at that age would not be a mental age match from my crude calculations
  • On the ERP component figures, it is recommended, if possible, to delineate more clearly the SHO1’s values as, understandably as it is a single individual, it is very much undershadowed by the descriptions of the young and old groups.
  • Line 294 – change ‘do’ to ‘does’ prior to have epilepsy
  • Line 295 – change ‘do’ to ‘does’ prior to have enough symptoms to get a diagnosis of autism
  • In materials and methods, there is the rationale for the age of the old group but not for the young

Author Response

Thank you for the opportunity to review this manuscript. The authors present a case study of an individual with a microduplication on the SHANK3 gene. These types of studies are important for deeply understanding the phenotype and how that can help us understand both underlying neurology and genetics of other disorders. The level of detail in this manuscript is nice and the addition of the EEG findings offers novel information. My biggest concern with the manuscript was in the presentation of the behavioral phenotype. These are detailed below. 

We appreciate the reviewer thoughtful comments and did our best to implement all suggestions provided. Corresponding changes with the reference to the line numbers in the corrected final ms. are listed together with detailed response to each comment.

Point 1. Provide slightly more detailed and referenced introduction

  • Lines 40-44: Where do the authors get the information on the prevalence and symtompatology. There are at least 2000 individuals with PMS in the worldwide registry and we know that that would be a large underestimate given the likely prevalence of 1:8000-15,000. Further, though the deletion may be difficult to detect genetically, there are lots of behavioral symptoms evidenced from an early age, typically infancy. At the very least these assertions should be referenced.

Response 1.1: We have corrected the information about prevalence and have provided detailed references to the articles from where the information about symptoms and prevalence of PMS was taken in lines 40-48

  • Line 62, I would recommend being more specific rather than saying ‘mental problems’
  • Response 1.2: We included more detailed information about mental disabilities of patients with 22q13 duplication as well as references to corresponding studies in lines 56-62.

Point 2. Reorganize and strengthen SH01 phenotype description

  • Section 2.2 – The diagnosis information is very vague. What are ‘other deficits of behavior due to unspecified causes’? This seems out of place with actual WHO diagnoses also provided. Where did the diagnoses come from? Did the research team do them or were they provided from past reports? The paragraph also becomes redundant with the parents complaining of intellectual disability and speech underdevelopment. Since the case study is the primary focus of this paper, I would recommend reworking so that this section is more flushed out.
  • We thank the reviewer for raising this issue and valuable suggestions. We included missing information and re-arranged the text not to confuse readers (e.g., F70.88 diagnosis reads as “mild mental retardation and other deficits of behavior due to other specified causes” with values after dot pointing to some additions to the major diagnosis of mental retardation). We also re-arranged paragraphs and sections. The paragraph on diagnosis also was extended by reports from psychologist, who performed IQ test.

See lines 165-178:

“Her official diagnoses were mild mental retardation and other deficits of behavior due to other specified causes (F70.88), and organic emotionally labile [asthenic] disorder with unspecified cause (F06.69). Diagnoses were obtained from the recent clinical reports provided by experienced psychiatrists from Moscow research institute of psychiatry and Scientific and practical center for mental health of children and adolescents, leading Moscow organization for diagnosis of mental health problems. The report from a psychologist confirmed mild mental retardation by Wechsler Intelligence Scale for Children (Russian adaptation based on original Wechsler Intelligence Scale for Children [70]): verbal IQ = 71; nonverbal/performance IQ = 64; full-scale IQ = 64. The psychologist also pointed unstable attention, smaller memory span, fluctuating but lower speed of performance, quicker tiredness and loss of work efficiency, as well as infantilism, protest behavior, irritability, emotional liability, problems with understanding the social context, lack of self-critique and motivation to overcome difficulties, preponderance of recreational entertaining interests over educational and cognitive ones”.

  • What is meant by ‘on examination she showed infantile behavior, too dependent on her mom’? I would also flush this out as a behavioral observation section so we can get a better picture of her as this also does not really align with her ability to succeed in a mainstream classroom

This sentence was moved into the section with current concerns and behavior at EEG/ERP examination and extended by the example.

Lines 181-192:

Parents’ major concerns at age 15 were learning disabilities, behavioral disorders, and irritability. The girl was sociable, and her mild cognitive impairment were hardly noticeable in daily routines and without special assessment in challenging/educational situation. Her mild speech underdevelopment manifested in rare problems to pronounce long and complex words and smaller vocabulary than typically developing (TD) peers. She attended the 9th grade of normal middle school that required great efforts from her parents. While she hardly managed to make any homework by herself, she was considering continuing her education in high school, pointing to the lack of adequate self-assessment. Among her interests was performing in a school theatre. She used her right hand to write and to eat. At EEG/ERP examination she showed infantile childish behavior, demanding attention of others and especially her mom, that was not typical for her age (e.g., she asked her mom to stay with her in experimental room).” 

  • Line 160, this paragraph also seems out of place. If the family history is a heading, I would denote it as such.

We removed the redundant information and called this section anamnesis. We also placed it at the beginning of the 2.2. Phenotyping. Clinical description.

  • To give context to the motor milestones, particularly because of noted low tone at least in those with the deletion, I would recommend starting with motor milestones were achieved within normal limits with holding her head at ….. This would give the reader context to the fact that these were within typical limits. Similarly, I might say that language milestones however were delayed with …
  • With respect to cognitive, I would then just say cognitive or precognitive milestones were also delayed, with lack of interest in books until and learning was also delayed (or whatever is meant by moderately retarded in mental development).
  • With regard to the aggression, I would say that it resolved or something else rather than eliminated but also would be nice to know if she had treatments for this.

We followed the reviewer’s suggestion. Currently the section reads as

Lines 149-164:

Anamnesis. SH01 was from full-term pregnancy from healthy parents, who were 39 years old at the time of the girl's birth. Her weight at birth was 3.040 g and length 52 cm, Apgar score was 7/8. Motor milestones were achieved within normal limits with holding her head at 2 months, sat down at 6 months, stood with the support at 10 months, began to walk alone at 11 months. Language milestones however were slightly delayed with the first syllables appeared at 12 months followed by the relatively long time of no phrases. Short sentences appeared at the age of 3. Cognitive development was also delayed, with lack of interest in books and cartoons until age 3. At about this age SH01 developed aggressiveness towards peers (e.g. biting) and protest behavior. At kindergarten she referred her bad behavior to fictional peer-boy. Aggressive behavior is resolved when she was about 10 years old. Currently she might have some rare periods of self-aggression (biting) when too angry and unsatisfied. SH01 started normal school together with TD peers but by the end of primary school she was referred to specialists due to the problems with dealing with school program (especially Math). However, she managed to continue the study in the school with TD peer with the support of the specialists and parents. Menstruation was regular and started at 10 years of age.”

  • For the ASD behaviors, was the SRS current? Similar with the ADI?

Yes, they were current. This information is currently included in the ms.

Line 197: “Autistic characteristics as assessed at age 15”

  • Line 176, assuming that the authors mean the Wechsler scale, though it would be good to know which version

We added the reference for the version that was used in lines 171-174:

“The report from a psychologist confirmed mild mental retardation by Wechsler Intelligence Scale for Children (Russian adaptation based on original Wechsler Intelligence Scale for Children [70]): verbal IQ = 71; nonverbal/performance IQ = 64; full-scale IQ = 64.”

Point 3 the rationale for the old and young comparison groups

  • Section 2.4 – what is the rationale for the old and young comparison groups? The young group at that age would not be a mental age match from my crude calculations
  • In materials and methods, there is the rationale for the age of the old group but not for the young

Response 3: In the section 2.4 we added justification for comparison with two groups of different ages. Our logic was to determine if our subject's ERP and ASSR results could be related to the maturity of the nervous system, and we relied on studies of changes in auditory evoked potentials and ASSR with age. We address to these studies in detail in the discussion section.

Lines 242-245:

“Comparison groups of different ages were selected to examine if the suggested alternation in SH01’s neurophysiological responses to sounds might be linked to the developmental delay in brain maturation as changes of AEP and ASSR with age are known (see discussion) or represent more general phenomena.”

Point 4 Figure

  • On the ERP component figures, it is recommended, if possible, to delineate more clearly the SHO1’s values as, understandably as it is a single individual, it is very much undershadowed by the descriptions of the young and old groups.

Response 4: we corrected figures describing ERP components with more understandable presentation of SH01’s values. 

Point 5 Spellings/grammar corrections

Response 5: We corrected all indicated grammar/spelling mistakes as well as others that we identified through re-reading of the ms.

  • Line 120, missing a ‘the’ before Rosetta Stone

done

  • Line 157, the ‘has’ is not necessary before ‘…started at 10 years of age’

done

  • Line 294 – change ‘do’ to ‘does’ prior to have epilepsy

done

  • Line 295 – change ‘do’ to ‘does’ prior to have enough symptoms to get a diagnosis of autism

done
